# SIR-ABSC: Incorporating Syntax into RoBERTa-based Sentiment Analysis Models with a Special Aggregator Token

**Ikhyun Cho** and **Yoonhwa Jung** and **Julia Hockenmaier**
University of Illinois at Urbana-Champaign
{ihcho2, yoonhwa2, juliahmr}@illinois.edu

## Abstract

We present a simple, but effective method to incorporate syntactic dependency information directly into transformer-based language models (e.g. RoBERTa) for tasks such as Aspect-Based Sentiment Classification (ABSC), where the desired output depends on specific input tokens. In contrast to prior approaches to ABSC that capture syntax by combining language models with graph neural networks over dependency trees, our model, Syntax-Integrated RoBERTa for ABSC (SIR-ABSC) incorporates syntax directly into the language model by using a novel aggregator token. SIR-ABSC[1] outperforms these more complex models, yielding new state-of-the-art results on ABSC.

## 1 Introduction

Aspect-Based Sentiment Classification (ABSC, Pontiki et al. (2014), Figure 1) is a fine-grained sentiment analysis task that aims to handle the fact that even simple statements such as *"The ambience was nice, but service wasn't so great."* may express different sentiments towards different aspects (this reviewer is positive about the restaurant's *"ambience"*, but negative about its *"service"*). In ABSC, the aspect to be classified is identified by a target string in the input sentence (e.g. *"ambience"*), and systems have to return the polarity (positive, neutral, negative) of the corresponding sentiment.

Pre-trained language models (PLMs) have been shown to work well for ABSC, presumably because their attention mechanisms capture semantic connections between target and context words (Li et al., 2019; Xu et al., 2020; Karimi et al., 2021). Starting with Do et al. (2019), PLMs have been supplemented with syntactic features, typically extracted from dependency graphs. This is typically done by using the word embeddings obtained from the PLM to initialize the node embeddings of a

graph neural network (GNN) obtained from the dependency graph (Tang et al., 2020; Wang et al., 2020; Hou et al., 2021; Xiao et al., 2022). However, such combined models have two major limitations:

1. **Suboptimal Interaction:** A typical challenge in combining PLMs and GNNs is to make the two models effectively interact with each other. Some approaches (Tang et al., 2020; Lu et al., 2020) attempt to accomplish this through heavy model architecture engineering. However, the PLM and GNN still operate in an asynchronous manner (one after the other), limiting their interaction, and yielding only a minor improvement in performance. We hypothesize that more integrated models can yield larger boosts in performance.

2. **Suboptimal Aggregation:** Existing (PLM + GNN) models cannot emphasize information at a specific distance(s) away from the target token. This is problematic since the models pre-define the number of GNN layers beforehand (Bai et al., 2020; Veyseh et al., 2020; Zhao et al., 2022). For example, if the number of GNN layers is fixed to three while an input sequence has the key sentiment word at a distance of one, the redundant second and third GNN layers can introduce noise. The ability to identify the most important distance(s) (one in this case) and to focus on that specific distance(s) based on the input is crucial to reduce such noise.

In order to alleviate these limitations, we propose syntax-integrated RoBERTa (SIR-ABSC), a novel framework for effectively augmenting PLMs with syntactic information. We chose RoBERTa (Liu et al., 2019) as our PLM baseline due to its notable performance on ABSC (Dai et al., 2021).

Instead of stacking multiple GNN layers on top of a PLM as in most previous works, we insert a

---

[1]The code for SIR-ABSC is publicly available at https://github.com/ihcho2/SIR-ABSC

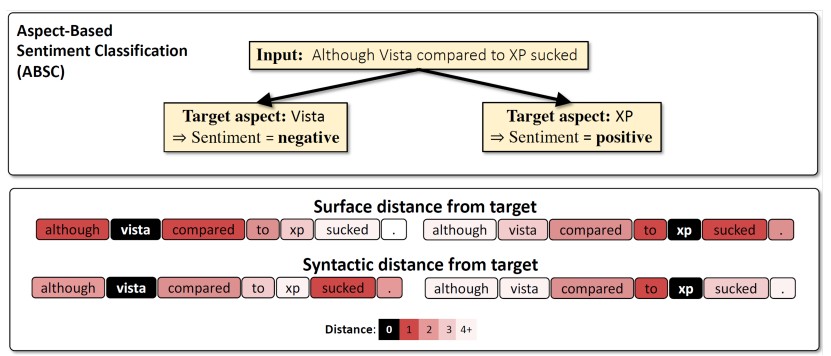
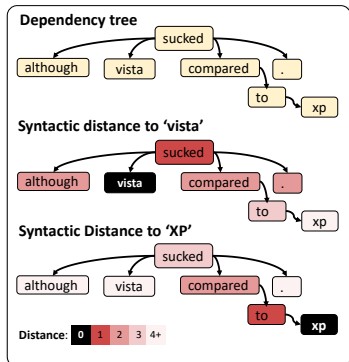

Figure 1: (Left Top) In ABSC, the sentiment to be predicted depends on the desired target aspect (words from the input). (Left Bottom) For ABSC, syntactic distance can be more informative than surface distance. (Right) Syntactic distances as defined by a dependency tree

**new [g] token** into the PLM that acts as a *syntax aggregator* and effectively replaces the work of GNN layers. A **Variable Distance Control (VDC)** mechanism allows [g] to focus on capturing syntactic knowledge by using constrained attention masks that reflect the graph structure. A **Dependency-Aware Aggregation (DAA)** mechanism leverages dependency label information. We further improve our model with **Automatic VDC Learning (Auto-VDC)**, which enables SIR-ABSC to capture and focus on important syntactic distances based on the input, an important advantage over GNNs. SIR-ABSC outperforms prior approaches, and establishes a new state of the art, on the most widely used ABSC datasets.

Our main contributions are summarized below :

- We present a novel approach to incorporating syntactic information into PLM through the use of a syntax aggregator token.

- We highlight two limitations in current GNN-based approaches (suboptimal interaction and aggregation) and present effective methods (DAA and Auto-VDC) to address them.

- To our knowledge, this is the first work to utilize dependency graph information *without* resorting to GNNs in ABSC.

- Our model achieves state-of-the-art results on two out of the four widely used ABSC datasets and demonstrates competitive performance on the remaining ones.

## 2 Aspect-Based Sentiment Classification

In ABSC, illustrated in Figure 1, the task is to predict the polarity (positive, negative or neutral) of the sentiment in input sentence $s = [w_1, w_2, ..., w_p, ..., w_{p+m-1}, ..., w_n]$ towards a given target aspect $t$ (a substring of the input sentence: $t_i = \{w_p, ..., w_{p+m-1}\}$).

### 2.1 Language Models for ABSC

Pre-trained language models (PLMs) such as BERT (Devlin et al., 2019) and RoBERTa (Liu et al., 2019) have gained predominance for many NLP tasks, including ABSC. RoBERTa, a variant of BERT, is known to show notable performance on ABSC tasks (Dai et al., 2021), and forms the basis of the models explored in this paper. RoBERTa (and BERT) are (pre)trained on large amounts of raw text with a masked language modeling objective. Both models use a Transformer (Vaswani et al., 2017) architecture in which each token's embedding is fed through multiple layers such that each token's embedding in a given layer can attend to all tokens in the sequence (in the same layer). To adapt these models for classification tasks, a special token ([CLS] for BERT, [s] for RoBERTa) whose output is fed into a task-specific feedforward layer is included in the input sequence. A separation token ([SEP] or [/s]) can be used to separate the input sequence from other task-specific information.

For the ABSC task, RoBERTa is typically used as follows: after tokenization, the input sentence is constructed as '[s] input sentence [/s] [/s] aspect sequence [/s]', where the aspect sequence includes the target aspect itself. The [s] token of the last layer is used for the final prediction and fine-tuning.

### 2.2 Combining PLMs with syntax

A common approach to ABSC is to supplement a PLM with syntactic information obtained from a

dependency parser. In a dependency graph (Figure 1) each word in the sentence corresponds to a node, with labeled edges indicating word-word dependencies. Note that the syntactic distance between words (e.g. *sucked* and *vista*) can be much smaller than their surface (string) distance (Figure 1).

Since the dependency parser and the PLM may use different tokenizers, tokenization needs to be broken into two stages to integrate both models seamlessly. The input sentence is first tokenized by the dependency parser, and then each token is again tokenized by RoBERTa's tokenizer, following previous work (Tang et al., 2020).

**Graph Neural Network-based ABSC models** To incorporate syntax into ABSC models, PLMs have been augmented with Graph Neural Networks (GNNs, Kipf and Welling (2016)) that capture the structure of the sentence's dependency tree. Although there are many variants (Trisna and Jie, 2022), the basic idea behind GNNs is to represent each node as a vector $h_i$ that is updated via graph convolution in each layer ($l \in [1, 2, \ldots L]$) of the GNN (Kipf and Welling, 2016) by aggregating its neighborhood information from the previous layer:

$$h_i^l = \sigma(A_{ij} W_l h_j^{l-1} + b_l)$$

$\sigma$ is an activation function, $W$ and $b$ are learnable parameters, and $A_{ij}$ is the entry of the graphs adjacency matrix that indicates whether nodes $i$ and $j$ are connected (in which case $A_{ij} = 1$; otherwise $A_{ij} = 0$). The general framework for augmenting PLMs with GNNs for ABSC is to initialize the GNN node embeddings with the PLM's output embeddings and use the final embeddings of the target aspects in the last GNN layer for classification.

Zhang et al. (2019) were the first to implement a GNN-based model for ABSC, adding a multi-layered Graph Convolutional Network (GCN) to encode dependency graphs on top of the word embedding layer. Wang et al. (2020) and Bai et al. (2020) proposed a relational graph attention network (R-GAT), which computes an additional attention distribution using the dependency label embeddings. Tang et al. (2020) strengthened interactions between contextual and graph representations through a mutual biaffine module. Mei et al. (2023) incorporated part-of-speech, distance, and syntactic dependency in a supervised manner. Several recent research in ABSC has tried to revise dependency graphs due to their imperfections or add external resources (Xiao et al., 2021, 2022; Liang et al.,

2022a). What is common to all these approaches is that the PLM and GNN operate in a serial fashion, and are not tightly integrated.

**Attention-mask based approaches** Another promising approach to incorporate syntactic information into PLMs that is more related to this paper, is to manipulate the Transformer's self-attention masks. For example, Syntax-BERT (Bai et al., 2021) uses multiple masks induced from the syntactic trees (e.g., parent, children, sibling, pairwise masks) to incorporate syntactic information into BERT. To do so, it requires multiple (usually more than 90) sub-networks, which share the same model parameters in a masked self-attention module, and then the outputs are aggregated through another attention layer, named a topical attention module.

The key difference between Syntax-BERT and SIR-ABSC is that Syntax-BERT alters all the input tokens' attention masks and aggregates them in a serial manner, while SIR-ABSC (which is specifically designed for tasks like ABSC, where the desired output depends on specific parts of the input) keeps the original input tokens intact while only modifying the attention-mask of the newly added [g] token.

## 3 SIR-ABSC

We present SIR-ABSC (Figure 2), which incorporates syntactic information into PLM designed for ABSC tasks. We accomplish this goal by augmenting RoBERTa with four components: (1) a single additional input token, named [g], whose attention masks depend on the structure of the input's dependency tree(s), paired with (2) a "variable distance control" mechanism that specifies how the structure of the dependency graph is reflected in [g]'s attention masks, (3) a dependency-aware aggregation which incorporates dependency labels using embeddings, and (4) an "Auto-VDC" mechanism that learns the ideal VDC for a given input.

**Input and output** After tokenization with RoBERTa's tokenizer, the input to SIR-ABSC is '[s] [g] input sentence [/s] [/s] aspect sequence [/s]', where the aspect sequence is the token "[g]" followed by the target aspect words. [g] uses its own independent dictionary embedding in the input layer but is initialized with the initial [s] embedding. We evaluate this choice in Section 5.

To obtain the output, the final layers of the [s] and [g] tokens are pooled and fed to a classification

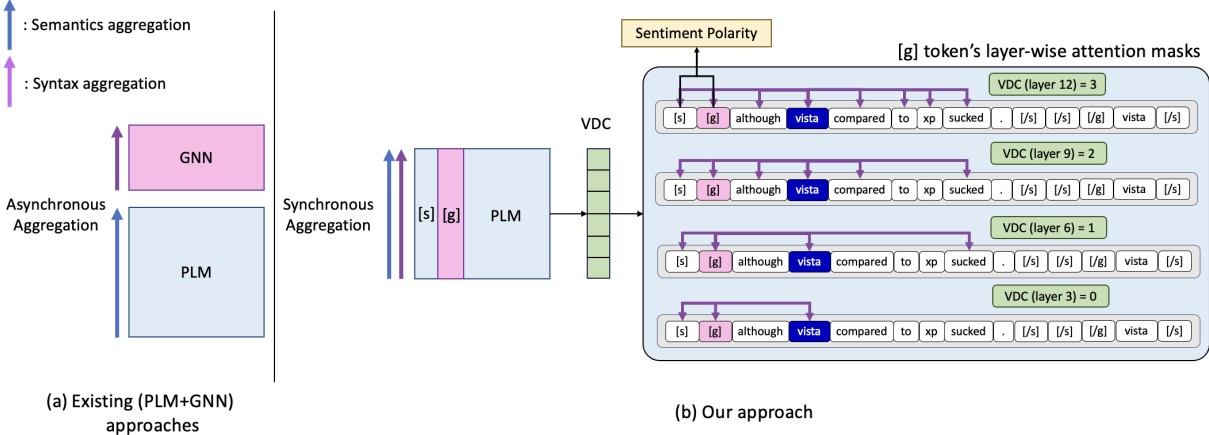

Figure 2: The overall architecture of our approach. Instead of stacking GNNs on top of a PLM (left), SIR-ABSC inserts a syntax aggregator token, [g], into the PLM for a synchronous and deeper aggregation (right). A heuristic VDC of [0,0,0,1,1,1,2,2,2,3,3,3] is used in this example ("vista" is the target aspect). DAA is not explicitly represented for the sake of simplicity. Best viewed in color.

layer as follows. First, a global vector $c$ is created by averaging [s] and [g] embeddings followed by a weight $W_c$ and an activation function $f$:

$$c = f\left(\frac{1}{2}(X_s + X_g)W_c\right)$$

Then, the final pooled vector ($h$) is:

$$h = \alpha_1 X_s + \alpha_2 X_g$$

where,

$$(\alpha_1, \alpha_2) = \text{softmax}(cX_s^T, cX_g^T)$$

Then, $h$ is fed into a fully connected layer ($W_{FC}$) to calculate the final sentiment polarity score $y \in \mathbb{R}^C$:

$$y = \text{softmax}(hW_{FC})$$

where $C$ is the number of sentiment polarity classes. We use the cross-entropy loss function as the objective function.

**The [g] token and distance-based attention masks** The [g] token is inserted next to the [s] token. Unlike the [s] token that attends to every token in the input, we allow each layer $l$ of [g] to only attend to the subset of input tokens $j$ that are at most a distance $D(j) \leq d_l$ away from the target aspect, allowing us to capture the intuition that the relevance of each word in a sentence to ABSC depends on its distance to the target aspect words. We achieve this with distance-based attention masks (Figure 2) that depend either on syntactic or surface distance, and are only used for the [g] token, and the VDC hyperparameters explained below that define the parameter $d_l$. We do not restrict how the input tokens can attend to [g].

For our baseline model, we assume that each layer $l$ of [g] is associated with one specific distance $d_l$. This yields a layer-specific attention mask $\mathbf{m}_g^{d=d_l}$ (a vector whose elements $\mathbf{m}_{g,j}^{d=d_l}$ are 0 if the distance $D(j)$ between token $j$ and the target aspect words is greater than $d_l$, and 1 otherwise). If distance is syntax-based, $D(j)$ is the length of the shortest path between token $j$ and the target aspect (so, if the target aspect consists of multiple tokens, we take the minimum distance to any of its component tokens). If the distance is surface-based, $D(j)$ is simply the token distance to the target aspect.

To summarize, all tokens except [g] are updated precisely as in the original PLM using the original attention mask $\mathbf{m}$, and [g] is updated using its own attention mask $\mathbf{m}_g^{d=d_l}$ as follows:

$$\begin{cases} X_{out,i\neq g}^l = \text{softmax}\left(\dfrac{Q_i^l K^{l^T}}{\sqrt{d}} + \mathbf{m}\right)V^l, & (1) \\[2em] X_{out,i=g}^l = \text{softmax}\left(\dfrac{Q_g^l K^{l^T}}{\sqrt{d}} + \mathbf{m}_g^{d=d_l}\right)V^l & (2) \end{cases}$$

where,

$$Q^l = X_{in}^l W_Q^l, K^l = X_{in}^l W_K^l, V^l = X_{in}^l W_V^l$$

$$m_{g,j}^{d=d_l} = \begin{cases} 0 & \text{, if } D(j) \leq d_l \\ -\inf & \text{, otherwise} \end{cases}$$

$X_{in}^l$ and $X_{out}^l$ are the input and output token embeddings of the $l$-th self-attention layer, $\mathbf{m}$ is the attention mask that cancels out padding tokens, $\mathbf{m}_g^{d=d_l}$ is the [g] token's attention mask, $Q^l, K^l, V^l$ are the $l$-th Query, Key, and Value weights, respectively,

$i$ is the token index, and $g$ is the [g] token's index. For simplicity, we ignore the relatively minor details, such as the FFN and normalization layers.

Unlike previous works where PLM and GNN operate one after the other, SIR-ABSC is a synchronous and fully-integrated model in that [s] (= semantic aggregator) and [g] (= syntactic aggregator) can attend to and affect each other's embeddings at every layer. We verify that this fully-integrated characteristic is crucial for SIR-ABSC's performance in Section 5.

**Variable Distance Control (VDC)**  To specify the attention masks used by the [g] token, we introduce a new set of hyper-parameters named Variable Distance Control (VDC). VDC is a list of 12 non-negative integers where the $l$-th element represents the $d_l$ value of the $l$-th layer of the [g] token: With a VDC of [0,0,0,0,0,0,1,1,1,1,1,1], the first six layers of the [g] token attend only to the target aspect, and the remaining six layers attend to tokens connected to the target via a direct dependency link.

Note that increasing VDCs (e.g., [0,0,0,0,1,1,1,1,2,2,2,2], [0,0,0,1,1,1,2,2,2,3,3,3]) can be used to mimic how GNNs work. Through graph convolution, the $i$-th layer of a GNN aggregates features of nodes up to length $i$ away from each node in the graph, allowing the GNN to gradually aggregate information from more and more distant nodes in its upper layers. Empirical results in Section 5 show that increasing VDCs have indeed better performance than constant VDCs (e.g., [2,2,2,2,2,2,2,2,2,2,2,2]) or decreasing VDCs (e.g. [3,3,3,2,2,2,1,1,1,0,0,0]). Here, our baseline model uses heuristically defined VDCs, referred to as "heuristic-VDC" in this paper.

**Dependency-aware aggregation (DAA)**  VDC allows the [g] token to distinguish tokens with different syntactic distances. However, the model is unaware of the dependency edge labels which could contain valuable information. To consider this information in SIR-ABSC, we re-formulate the update of the [g] token (Eq. 2) by including learnable embedding vectors corresponding to the dependency label to each token as follows:

$$X_{out,i=g}^l = \text{softmax}\left(\frac{Q_g^l \tilde{K}^{l^T}}{\sqrt{d}} + \mathbf{m}_g^{d=d_l}\right)\tilde{V}^l$$

where $X_{in}^l$, $K^l$ and $V^l$ in equation 2 are modified to $\tilde{X}_{in}^l$, $\tilde{K}^l$ and $\tilde{V}^l$ to consider the dependency in-

formation as below:

$$\tilde{X}_{in}^l = f(X_{in}^l + X_{dep}W_{dep}^l)$$
$$\tilde{K}^l = \tilde{X}_{in}^l W_K, \tilde{V}^l = \tilde{X}_{in}^l W_V$$

$f$ is the activation function, $X_{dep}$ is the dependency label embedding of the input tokens, $W_{dep}^l$ is a learnable weight matrix that maps dependency label embeddings to the size of the PLM's embedding. This allows SIR-ABSC to leverage the dependency label information for better aggregation.

**Automatic VDC learning (Auto-VDC)**  In Section 5, Table 2 shows the significance of the VDC configuration in SIR-ABSC, with up to 1.1% difference in model performance. This result naturally gives rise to a challenging question: *How can we decide the ideal VDC configuration?*

Given numerous possible configurations for VDC, it is inefficient to rely on heuristics such as grid search. We therefore define a learning-based approach to finding the ideal VDC configuration by computing multiple [g] embeddings based on different VDC values and then merging them into one using a learnable weighted sum.

We first create multiple [g] candidates $X_{\{g,i\}}^l$'s where each uses the attention mask of VDC=$i$ as follows. First, the $Q_g^l$ vector is transformed to $Q_{\{g,i\}}^l$ by a weight $W_i^l$ and an activation $f$:

$$Q_{\{g,i\}}^l = f(Q_g^l W_i^l)$$

Then, $X_{\{g,i\}}^l$ is generated using attention mask of VDC of $i$:

$$X_{\{g,i\}}^l = \text{softmax}\left(\frac{Q_{\{g,i\}}^l \tilde{K}^{l^T}}{\sqrt{d}} + m_g^{d=i}\right)\tilde{V}^l$$

Then, we use a learnable weighted sum to create our final [g] vector $X_g^l$:

$$X_g^l = \alpha_0^l \cdot X_{\{g,0\}}^l + \cdots + \alpha_k^l \cdot X_{\{g,k\}}^l$$

where the weights are computed as follows. First, a vector $c$ is computed by averaging $X_{\{g,i\}}^l$ followed by a weight $W^l$ and an activation function $f$:

$$c = f\left(\frac{1}{k+1}\Big(\sum_{i=0}^{k} X_{\{g,i\}}^l\Big)W^l\right)$$

Then, the weights are computed by applying the softmax over the dot products with $c$:

$$(\alpha_0^l, ..., \alpha_k^l) = \text{softmax}(X_{\{g,0\}}^l c^T, ..., X_{\{g,k\}}^l c^T)$$

Auto-VDC is illustrated in Figures 3 and 4. this approach has the following advantages: First, due to the attention pooling, the ideal VDC of each layer can be automatically learned by the attention scores. That is, $X^l_{\{g,n\}}$ with the ideal VDC $= n$ for layer $l$ will be automatically assigned a higher weight. Furthermore, unlike heuristic-VDC, where the model must attend tokens up to a distance of $\max(\text{VDC})$, Auto-VDC can adjust which distances to focus on based on the input, thereby reducing the noise from the redundant distances.

## 4 Experimental Results

**Datasets and Experimental Settings** We use the most widely used ABSC data sets: the Laptop and Restaurant datasets from SemEval-2014 task 4 (Pontiki et al., 2014), the Twitter dataset of Dong et al. (2014), and the MAMS-ATSA dataset (Jiang et al., 2019). Table 5 in Appendix A shows the statistics of the ABSC datasets. For SIR-ABSC, we use the pre-trained RoBERTa-base model provided by huggingface. We use spaCy[2]'s en_core_web_sm model version 3.3.0 as the dependency parser. Finetuning uses a batch size of 32, dropout rate of 0.1, learning rate of {1.0e-5, 1.5e-5, 2.0e-5} using the AdamW optimizer, and 300 for the dependency embedding dimension. We run the experiments with five random seeds and report average accuracy and macro-F1. We used the same set of random seeds for each ablation experiment. All experiments are conducted on a single Tesla A100 GPU.

**Baselines** We compare SIR-ABSC with previous (PLM+Dependency graph) models in Table 1. Specifically, (1) DGEDT-BERT (Tang et al., 2020) proposes a biaffine module that deepens the interaction of semantic and syntactic representations; (2) kumaGCN (Chen et al., 2020) utilizes dependency graph along with a latent graph induced from self-attention neural networks; (3) dotGCN (Chen et al., 2022) uses GNN over an induced tree trained by reinforcement learning; (4) CHGMAN (Niu et al., 2022) uses a three-channel multi-view learning model on the dependency graph for better representation learning; (5) R-GAT (Wang et al., 2020) applies relational graph attention networks over the dependency graph to incorporate edge label information into the model; (6) DGNN (Xiao et al., 2022) uses dependency graph and an additional

adjacency matrix generated based on the syntactic distance from the target; (7) MWM-GCN (Zhao et al., 2022) uses masking based on the syntactic distance from the target and additional multi-head self-attention layers for better performance; (8) SG-GCN (Veyseh et al., 2020) applies a regulation gate on the tokens based on their similarity to the target; (9) RoBERTa-RGAT, PWCN (Dai et al., 2021) reproduces the baseline RoBERTa model with using dependency graph in two different approaches (relational GAT and point-wise convolution network); (10) AM-RoBERTa (Feng et al., 2022b) applies target-relevant masking to the self-attention mechanism in the PLM to focus more on target-related contexts; (11) RoBERTa-DLGM (Mei et al., 2023) proposes a supervision-based approach to incorporate syntax information into PLM.

Note that baselines that use external knowledge sources in addition to dependency graphs (e.g., Sentic GCN-BERT (Liang et al., 2022b), KGAN (Zhong et al., 2023), SGAN (Yuan et al., 2022)) are not direct competitors to SIR-ABSC, although SIR-ABSC still outperforms all these methods except for KGAN on Twitter (which is not a fair comparison since KGAN relies on an external knowledge source, WordNet). BERT4GCN (Xiao et al., 2021) is excluded since it uses a different experiment setting while no public code is available. It is also worth noting that all these baseline models use GNN-based modules to aggregate syntactic information. SIR-ABSC, on the other hand, presents a new aggregating approach via the syntax aggregator token without resorting to GNNs.

**Overall Results** Table 1 compares SIR-ABSC against all competitive RoBERTa+GNN or BERT+GNN combination models that use dependency graphs extracted from parsers [2][3][4] (Tang et al., 2020; Wang et al., 2020; Xiao et al., 2022; Zhao et al., 2022; Liang et al., 2022a; Veyseh et al., 2020; Xiao et al., 2021; Dai et al., 2021; Feng et al., 2022a; Mei et al., 2023). Notably, SIR-ABSC outperforms all previous models on Laptop14, Restaurant14, and MAMS-ATSA datasets, establishing a new state-of-the-art record. On Twitter, SIR-ABSC clearly outperforms RoBERTa-based models and is competitive with the (overall better performing) BERT-based models. Signifi-

---

[2]spaCy parsers: https://spacy.io/

[3]Stanford CoreNLP: https://stanfordnlp.github.io/CoreNLP/

[4]Biaffine Parser (Dozat and Manning, 2016) from the AllenNLP package https://allenai.org/allennlp

Figure 3: With Auto-VDC, the [g] token automatically learns the ideal VDC configuration. For example, in the lower layers, the [g] token's attention scores focus on the target aspect ("touch screen functions") and the [s] token. As it moves to the upper layers, the [g] token shifts its focus to the critical phrase "did not enjoy" for the final prediction. This shows that Auto-VDC progressively consolidates information from increasingly distant nodes in the upper layers, just like heuristic VDCs and GNNs, but in an automatic way, thereby diminishing the need for manual heuristic adjustments.

| Base PLM | Models | Syntax | External Resource | Lap14 | | Rest14 | | Twitter | | MAMs | |
|---|---|---|---|---|---|---|---|---|---|---|---|
| | | | | Acc. | F1 | Acc. | F1 | Acc. | F1 | Acc. | F1 |
| BERT | DGEDT-BERT[2] | Dep. Graph | x | 79.8 | 75.6 | 86.3 | 80.0 | 77.9 | 75.4 | – | – |
| | kumaGCN | Dep. Graph | x | 82.0 | 78.8 | 86.4 | 80.3 | 77.9 | 77.0 | – | – |
| | dotGCN | x | x | 81.0 | 78.1 | 86.2 | 80.5 | 78.1 | 77.0 | 85.0 | 84.4 |
| | CHGMAN | Dep. Graph | x | 81.5 | 77.7 | 86.9 | 81.6 | – | – | **85.1** | 84.3 |
| | RGAT-BERT[4] | Dep. Graph | x | 78.2 | 74.1 | 86.6 | 81.4 | 76.2 | 74.9 | – | – |
| | DGNN (BERT)[4] | Dep. Graph | x | 81.4 | 79.0 | 87.2 | 81.7 | 76.2 | 75.0 | – | – |
| | MWM-GCN (BERT)[3] | Dep. Graph | x | 82.8 | 79.5 | 88.5 | 82.6 | 78.9 | 77.4 | – | – |
| | Sentic GCN-BERT[2] | Dep. Graph | ✓(SenticNet) | 82.1 | 79.1 | 86.9 | 81.0 | – | – | – | – |
| | SGGCN-BERT | Dep. Graph | x | 82.8 | 80.2 | 87.2 | 82.5 | – | – | – | – |
| | **Ours: SIR-ABSC (BERT)** [2] | Dep. Graph | x | 82.2(±0.4) | 79.1(±0.4) | 87.3(±0.7) | 81.6(±0.8) | 77.7(±0.8) | 76.6(±0.7) | 84.6(±0.4) | 83.3(±0.5) |
| RoBERTa | RoBERTa-RGAT[4] | Dep. Graph | x | 83.4 | 80.3 | 87.4 | 80.6 | 74.4 | 72.9 | 84.5 | 83.7 |
| | RoBERTa-PWCN[2] | Dep. Graph | x | 84.2 | 81.2 | 87.4 | 81.1 | 76.6 | 75.6 | – | – |
| | AM-RoBERTa[2] | x | x | 83.0 | 80.2 | 88.1 | 82.5 | – | – | 84.2 | 83.7 |
| | RoBERTa-DLGM[2] | Dep. Graph | x | 84.4 | 82.0 | 88.6 | 83.6 | 75.5 | 74.6 | 84.8 | 84.3 |
| | KGAN | Dep. Graph | ✓(WordNet) | 83.9 | 81.1 | 88.5 | 84.1 | **80.6** | **79.6** | – | – |
| | **Ours: SIR-ABSC** [2] | Dep. Graph | x | **85.0(±0.5)** | **82.1(±0.6)** | **89.7(±0.6)** | **84.8(±0.9)** | 77.5(±0.6) | 76.4(±0.6) | 85.0(±0.3) | **84.5(±0.3)** |

Table 1: SIR-ABSC outperforms all prior works on Laptop and Restaurant, and is competitive on Twitter and MAMS.

cance tests following Dror et al. (2019) result in minimum epsilon values of 0 using $p$-value of $p < 0.01$, indicating that SIR-ABSC's performance is "stochastically greater" than the baselines (details in Appendix E). We also conducted experiments using BERT (SIR-ABSC (BERT)) instead of RoBERTa and it shows comparable performance with the state-of-the-art BERT models (details in Appendix G).

**Twitter and multi-sentence items**  Table 6 in Appendix A shows that the Twitter dataset has a large proportion of multi-sentence items. Since multi-sentence items have multiple dependency graphs, it requires us to combine them by adding a dummy root node that links to the heads of each sentence. This, as well as RoBERTA's generally lower performance on Twitter, maybe one reason why we do not achieve state-of-the-art on Twitter. We have also not attempted to examine how parser accuracy contributes to performance differences.

## 5 Analysis

We now examine the effect of the design decisions that distinguish SIR-ABSC from RoBERTa.

**Does [g] require syntactic distances?**  To understand the impact of syntax on SIR-ABSC, we now compare it to a variant that uses surface distance instead of syntactic distance. The surface (or position) distance of a token is computed simply by the number of tokens between the closest target aspect token and the corresponding token following previous works (Zeng et al., 2019; Phan and Ogunbona, 2020). Focusing on words near the target aspect is known to be effective in the ABSC task (Zeng et al., 2019). But syntactic distance is often very different from surface distance (see Figures 1 and 2.)

Table 2 shows results for all three VDC types (decreasing, constant, and increasing) under both metrics that indicate that syntactic distances yield generally better performance than surface distances, especially in the increasing VDC configuration.

| Variable Distance Control (VDC) | Lap14 | | Rest14 | | Twitter | | MAMs (Val.) | |
|---|---|---|---|---|---|---|---|---|
| | Acc. | F1 | Acc. | F1 | Acc. | F1 | Acc. | F1 |
| RoBERTa-ASC | 82.1(±0.9) | 78.9(±1.1) | 87.6(±0.8) | 81.7(±0.8) | 75.6(±0.6) | 74.5(±0.7) | 84.4(±0.4) | 83.9(±0.5) |
| SIR-ABSC *(Position Distance)* | | | | | | | | |
| • Decreasing-VDC | 83.4(±0.7) | 80.4(±0.8) | 88.5(±0.6) | 83.2(±0.6) | 76.5(±0.8) | 75.3(±0.7) | 84.7(±0.4) | 84.3(±0.4) |
| • Constant-VDC | 83.7(±0.7) | 80.7(±0.7) | 88.6(±0.6) | 83.3(±0.5) | 76.4(±0.7) | 75.4(±0.8) | 84.9(±0.4) | 84.4(±0.5) |
| • Increasing-VDC | 83.7(±0.9) | 80.5(±0.8) | 88.6(±0.7) | 83.2(±0.7) | 76.9(±0.9) | 76.0(±1.0) | 84.8(±0.3) | 84.4(±0.2) |
| SIR-ABSC *(Dependency Graph)* | | | | | | | | |
| • Decreasing-VDC | 83.8(±0.6) | 80.8(±0.7) | 88.2(±0.6) | 82.9(±0.6) | 76.5(±0.7) | 75.4(±0.8) | 84.8(±0.4) | 84.3(±0.4) |
| • Constant-VDC | 83.9(±0.6) | 80.7(±0.6) | 88.9(±0.7) | 83.7(±0.8) | 76.4(±0.8) | 75.2(±0.7) | 84.8(±0.4) | 84.5(±0.3) |
| • Increasing-VDC | **84.1(±0.8)** | **81.1(±0.8)** | **89.3(±0.6)** | **84.1(±0.8)** | **77.2(±0.9)** | **76.3(±0.9)** | **85.0(±0.3)** | **84.6(±0.3)** |

Table 2: Empirical results on the effectiveness of VDC. The results show that SIR-ABSC generally shows better performance in the order of decreasing < fixed < increasing VDCs. This result matches our intuition of [g] imitating GNN as described in Section 3. A more detailed result table is in the Appendix C.

| SIR-ABSC | | Lap14 | | Rest14 | | Twitter | | MAMs (Val) | |
|---|---|---|---|---|---|---|---|---|---|
| Auto-VDC | DAA | Acc. | F1 | Acc. | F1 | Acc. | F1 | Acc. | F1 |
| ✓ | ✓ | **85.0(±0.5)** | **82.1(±0.6)** | **89.7(±0.6)** | **84.8(±0.9)** | **77.5(±0.6)** | **76.4(±0.6)** | **85.4(±0.4)** | **84.9(±0.4)** |
| ✓ | x | 84.7(±0.6) | 81.9(±0.7) | 89.5(±0.5) | 84.4(±0.7) | 77.3(±0.6) | 76.2(±0.7) | 85.2(±0.3) | 84.8(±0.2) |
| x | ✓ | 84.6(±0.8) | 81.8(±0.8) | 89.4(±0.4) | 84.4(±0.8) | 77.2(±0.6) | 76.3(±0.6) | 85.2(±0.5) | 84.7(±0.5) |
| x | x | 84.1(±0.8) | 81.1(±0.8) | 89.3(±0.6) | 84.1(±0.8) | 77.2(±0.9) | 76.3(±0.9) | 85.0(±0.3) | 84.6(±0.3) |

Table 3: Ablation study results. The first row is equivalent to our final SIR-ABSC model and the last row is equivalent to our baseline SIR-ABSC model using increasing heuristic-VDC.

**Consistency with general GNNs** SIR-ABSC is inspired by how GNNs gradually aggregate information from nodes that are more and more distant in their upper layers. As mentioned in section 3, increasing VDC hyperparameters can be used to mimic this behavior. As mentioned above, Table 2 summarizes experiments conducted on three different types of VDCs: increasing (e.g., [0,0,0,1,1,1,2,2,2,3,3,3]), constant (e.g., [2,2,2,2,2,2,2,2,2,2,2,2]), and decreasing (e.g., [3,3,3,2,2,2,1,1,1,0,0,0]).

From the table, we can observe the followings. First, the VDC hyperparameter is crucial for SIR-ABSC's performance, verifying [g]'s effectiveness. Also, it can be seen that SIR-ABSC has the highest performance with increasing VDCs (i.e. when it is most similar to typical GNNs), and the lowest performance with decreasing VDCs (i.e. when it is the least similar to GNNs). More detailed experiment results are provided in Appendix C.

**How does SIR-ABSC alleviate the *suboptimal interaction* issue?** Existing (PLM+GNN) models in ABSC work in an asynchronous fashion (i.e., one after the other), which results in a weak interaction between the semantic and syntactic information (Tang et al., 2020). SIR-ABSC, on the other hand, is a fully-integrated model where [s] (=semantic aggregator) and [g] (=syntactic aggregator) interact with each other in every layer. To verify the strength of this fully-integrated characteristic, we conducted an ablation study (Table 4) where we intentionally block the attention masks between [s] and [g] in every layer. Consequently, the [s] and [g] tokens cannot attend to each other during the self-attention layers, weakening their interaction. By comparing SIR-ABSC with Table 4 Variant 1 (w/o s↔g interaction), we can observe that the drop in performance is considerable, verifying our hypothesis that more integrated models can alleviate the *suboptimal interaction* issue and yield more significant performance improvements.

**How does SIR-ABSC alleviate the *suboptimal aggregation* issue?** We conducted an ablation study to verify the effectiveness of DAA and Auto-VDC, and the results are summarized in Table 3. From the table, we can observe that each method improves the performance over our baseline model (using increasing heuristic-VDC), and both applied together further enhances the performance. These results verify that SIR-ABSC can effectively utilize the dependency labels and learns to identify and focus on specific distances based on the input, alleviating the *suboptimal aggregation* issue. We also include two case studies, Figures 3 and 4, for qualitative analysis. In Figure 3, SIR-ABSC's [g] focuses on the target aspect ("touchscreen functions") in the early layers and then gradually captures the key phrase "did not enjoy" in the later layers, showing some consistency with increasing heuristic-VDC results. Also, from Figure 4 we can see that SIR-ABSC learns to focus more on VDCs of 0,1, and 2 and less on 3 (see Figure 4

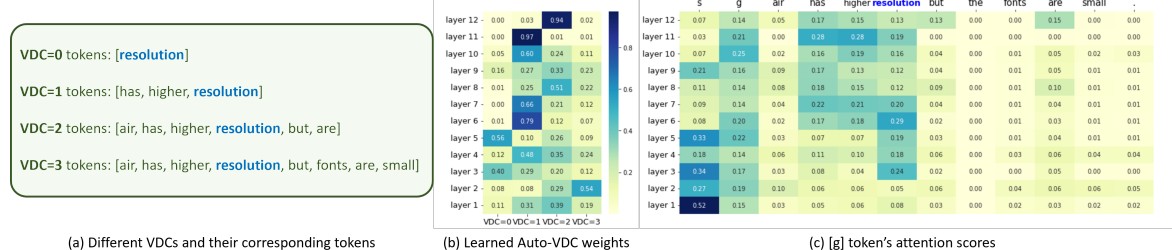

| | (a) Different VDCs and their corresponding tokens | (b) Learned Auto-VDC weights | (c) [g] token's attention scores |

Figure 4: A case study on the effect of the Auto-VDC mechanism. Target aspect is "resolution". Auto-VDC learns to focus more on the key distances (VDCs of 1 and 2) and less on the redundant distance (VDC of 3). As a result, the model successfully focuses on the key phrase "has higher resolution" at the last layers.

| [g] token | Lap14 | | Rest14 | | Twitter | | MAMs | |
|---|---|---|---|---|---|---|---|---|
| | Acc. | F1 | Acc. | F1 | Acc. | F1 | Acc. | F1 |
| SIR-ABSC | 85.0(±0.8) | 82.1(±0.9) | 89.7(±0.6) | 84.8(±0.9) | 77.5(±0.6) | 76.4(±0.6) | 85.2(±0.4) | 84.8(±0.4) |
| Variants | | | | | | | | |
| 1. w/o s ↔ g interaction | 84.2(±0.8) | 81.5(±0.9) | 89.3(±0.4) | 84.1(±0.6) | 77.2(±0.9) | 76.1(±0.9) | 85.0(±0.3) | 84.5(±0.3) |
| 2. [g] init. = aspect embed. | 84.7(±0.8) | 81.6(±0.7) | 89.3(±0.6) | 84.3(±0.6) | 77.2(±0.8) | 76.0(±0.9) | 85.0(±0.4) | 84.6(±0.4) |
| 3. SIR-ABSC-[g] (Appendix F) | 83.5(±0.6) | 80.5(±0.6) | 88.3(±0.4) | 82.9(±0.4) | 75.6(±0.8) | 74.3(±0.8) | 84.6(±0.3) | 84.2(±0.3) |

Table 4: Empirical results on the fully-integratedness and the inherent strength of the pre-trained [s] token embedding.

b) where VDC=3 tokens are indeed redundant (includes "fonts are small" where the target is "resolution"), verifying that SIR-ABSC can capture important distances based on the input. We provide more detailed ablation and case study results in Appendix D.

**Inherent strength of the pre-trained [s] token as an aggregator** There seems to be an inherent advantage in using the pre-trained embedding of the [s] token also for [g]. Table 4 (Variant 2) compares SIR-ABSC (in which the dictionary embedding of [g] is initialized with [s]'s embedding), with a variant in which we use the actual aspect word's dictionary embeddings as the [g] embedding (if the aspect consists of several words, we average their embeddings). Initializing [g] with the [s] embedding yields better performance, perhaps because the [s] embedding is better suited to aggregate information than the embeddings of other tokens, providing a better starting point for a sequence element that is also intended to aggregate information (albeit of a slightly different nature). Erase later: (Chen et al., 2020), (Chen et al., 2022), (Niu et al., 2022), (Wang et al., 2020), (Xiao et al., 2022), (Zhao et al., 2022), (Liang et al., 2022b), (Veyseh et al., 2020), (Dai et al., 2021), (Feng et al., 2022b), (Mei et al., 2023), (Xiao et al., 2021), (Liang et al., 2022a), (Yuan et al., 2022), (Zhong et al., 2023)

## 6 Conclusion

This paper has proposed a novel framework, SIR-ABSC, that effectively incorporates syntactic information directly into a pre-trained large language model (PLM) such as RoBERTa for tasks like Aspect-Based Sentiment Classification (ABSC), in which the desired output depends on specific words in the input, and where syntactic distance to the relevant input words may be important. In contrast to prior work, where a separate GNN was added to the output of the PLM, in our model, attention masks for new [g] token capture syntactic information, and a new hyper-parameter, named variable distance control (VDC), captures graph structure in a similar fashion. Dependency-aware aggregation (DAA) mechanism allows SIR-ABSC to use the dependency labels effectively, and finally, the Auto-VDC mechanism learns a mixture of multiple distances, allowing the model to identify and focus on important syntactic distances for a given input.

To the best of our knowledge, SIR-ABSC is the first model to incorporate syntactic knowledge into RoBERTa for ABSC *without* resorting to GNNs. Experiments show we achieve state-of-the-art performance in SemEval-2014 task 4. This demonstrates the efficiency of our approach and suggests a new paradigm for augmenting PLM with syntax.

## Limitations

In this section, we summarize the limitations of our model that can be improved in the future. First, we

did not explore various types of dependency parsers as mentioned in Section 4, which could result in different performances. Similarly, we did not examine how accurate the parser we used was and how much performance difference it could make, as mentioned in Section 4. Second, we did not consider other types of syntactic information besides dependency trees, such as phrase dependency graphs, which could also be effective for ABSC. Lastly, our method might be limited to single-sentence items. Although we inserted a dummy root node to merge multiple dependency graphs within a single input sentence, our model does not achieve state-of-the-art results on the Twitter dataset, which consists primarily of multi-sentence items. Therefore, it would be crucial to investigate more effective methods for merging multiple dependency graphs in a given input sentence.

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

# A  Details on datasets

Our model SIR-ABSC is evaluated on four different datasets from SemEval-2014 Task 4, Twitter, and MAMS datasets. Table 5 shows the statistics of the datasets. Table 6 shows the percentage of multiple sentences per item and the average number of sentences per item on each dataset.

| Dataset | Train | Test | Val |
|---|---|---|---|
| Restaurant (SemEval-2014) | 3608 | 1120 | - |
| Laptop (SemEval-2014) | 2328 | 638 | - |
| Twitter | 6248 | 692 | - |
| MAMS-ATSA | 11186 | 1336 | 1332 |

Table 5: Dataset Overview

| Datasets | Distribution | Train | Test | Val |
|---|---|---|---|---|
| Lap14 | % of multiple sent./item | 7.86 | 7.84 | - |
| | Avg. sent./item | 1.09 | 1.09 | - |
| Res14 | % of multiple sent./item | 4.02 | 4.38 | - |
| | Avg. number of sent./item | 1.04 | 1.05 | - |
| Twitter | % of multiple sent./item | 59.44 | 60.55 | - |
| | Avg. number of sent./item | 1.99 | 1.96 | - |
| MAMS-ATSA | % of multiple sent./item | 2.12 | 0.82 | 0.60 |
| | Avg. number of sent./item | 1.02 | 1.01 | 1.01 |

Table 6: Prevalence of multi-sentence items in the ABSC datasets.

# B  Comparing different pooler types

The [s] and [g] token outputs are combined after the last layer of SIR-ABSC encoders as described in Section 3. We conduct experiments on three different types of poolers for combining [s] and [g] token embeddings at the final layer: average, max, and attention pooling. Table 7 summarizes the results of using different pooler types for SIR-ABSC. The result shows that attention pooling shows better results in general.

# C  Detailed variable distance control results

Our variable distance control (VDC) is a unique hyper-parameter which consists of 12 non-negative integers, where each integer represents the $d_l$ value of the $l$-th layer. Theoretically there are exponentially many possible values for VDC but we use three representative types: increasing, constant, and decreasing VDCs.

We heuristically chose specific values for each type of VDCs and the detailed results are summarized in Table 8. The table shows that SIR-ABSC

| Pooler types | Lap14 | | Rest14 | | Twitter | |
|---|---|---|---|---|---|---|
| | Acc. | F1 | Acc. | F1 | Acc. | F1 |
| SIR-ABSC | | | | | | |
| w/ max pooling | 83.2 | 80.0 | 88.7 | 83.3 | 74.8 | 73.7 |
| w/ avg pooling | 83.8 | 80.6 | 88.8 | 83.5 | 76.5 | 75.5 |
| w/ att pooling | **84.1** | **81.1** | **89.3** | **84.1** | **77.2** | **76.3** |

Table 7: Comparing different pooler types for SIR-ABSC. We used VDC = [0,0,0,1,1,1,2,2,2,3,3,3] with the default full-interaction for the experiment.

has the highest performance with increasing VDCs. Increasing VDCs are designed to work as the most similar to the typical GNN by aggregating information from the closest nodes to farther nodes based on the target aspect. On the other hand, decreasing VDCs has the lowest performance where the decreasing VDCs are designed to work as least similar to a GNN in the opposite order (i.e., aggregating information from farther nodes to closer nodes based on the target aspect). From these results, we can conclude that SIR-ABSC successfully imitates the typical GNN mechanism through increasing VDC configuration.

## D Detailed ablation and case study results on DAA

We provide a more detailed ablation and case study results regarding DAA in Table 9 and Figure 5. We experiment two additional variants of the proposed DAA mechanism: DAA (w/o edge label) and DAA (w/o edge direction), to verify the effectiveness of different kinds of syntactic information. Specifically, DAA (w/o edge direction) ignores the direction of the edges and assign the embeddings solely based on the dependency label. DAA (w/o edge label) ignores the edge labels (e.g., nsubj) and assign the embeddings solely based on the edge direction. By comparing #1 and #2 from Table 9, we can observe that employing the VDC mechanism with the [g] token to replace GNNs leads to significant improvement in performance (average accuracy gain of 1.475% and average f1-score gain of 1.775). By further supplementing the model with missing information, such as edge direction and edge label information, DAA yields additional average gains of 0.2% on accuracy and 0.275 on f1-score over [RoBERTa+VDC] (#2 vs #5). The DAA ablation results (#3 and #4) show that the performance decreases for both DAA (w/o edge direction) and DAA (w/o edge label), indicating their importance in DAA performance. Consistent with our intuition,

we can observe that the edge label information is relatively more important than edge direction information in DAA. Another interesting finding is that DAA is more effective on BERT than RoBERTa. We speculate that this is because RoBERTa already utilizes the information contained in the dependency edge label to some extent, which is probably one of the reasons why RoBERTa shows better overall baseline performance than BERT.

Additionally, we incorporate two case studies to illustrate the effectiveness of DAA in Figure 5. From the above examples, we can observe that the attention scores of words (pretentious, and, inappropriate) and (anywhere, else) get much higher when DAA is applied. Our speculation is that this occurs because DAA learns the significance of "advcl" and "advmod" labels, resulting in high attention scores being assigned to expressions such as (pretentious $\xrightarrow{advcl}$ claim, be $\xrightarrow{advmod}$ else, and else $\xrightarrow{advcl}$ anywhere) thereby leading to correct predictions. This is also consistent with our intuition since adverbial modifiers are related to adjectives which is highly likely to be important for deciding the sentiment.

## E Significance tests

Following the proposed significance test designed for comparing deep learning models from (Dror et al., 2019), we ran their official code with our SIR-ABSC and the baseline models. Specifically, we compared (1) SIR-ABSC with RoBERTa-baseline and (2) SIR-ABSC (BERT) with BERT-baseline.

This test takes three components (results from method A, results from method B, and a confidence level) as input, and the purpose of this test is to answer the question: "Is A **almost stochastically greater** than B?" This test is based on the relatively new concept of "Almost Stochastic Order" which is a relaxed version of "Stochastic Order". The test outputs a minimum $\epsilon$ value where the result is interpreted as follows:

- If $\epsilon > 0.5$: We can't say A is almost stochastically greater than B.

- If $\epsilon \leq 0.5$: We can say A is almost stochastically greater than B.

- If $\epsilon = 0$: We can say A is "stochastically greater" than B not just "almost stochastically greater" which is the best result we can get in this test. Mathematically meaning $\inf\{x : t \leq F(x)\} \geq \inf\{x : t \leq G(x)\}$ for $\forall t \in (0, 1)$,

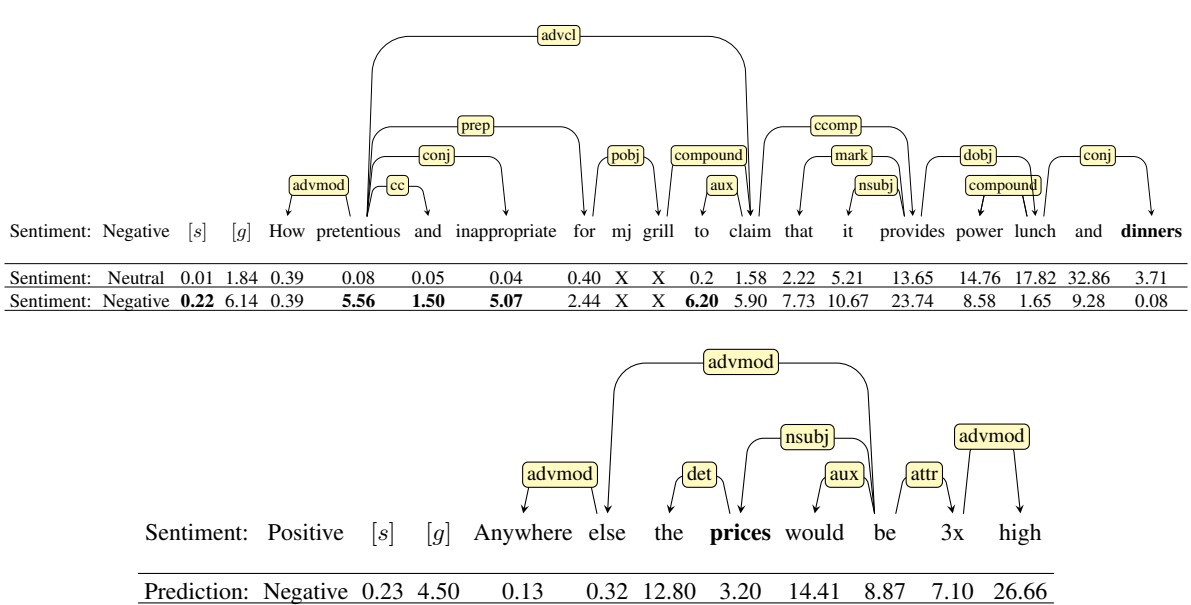

Figure 5: Two case studies employing DAA. The targets (dinners, prices) are marked in bold. The first and second rows below the sentence depict the attention scores of the [g] token without and with DAA, respectively. Attention scores that increase more than ten-fold when DAA is applied are also marked in bold. In these examples, DAA enables the [g] token to place greater emphasis on critical sentiment words like "pretentious," "inappropriate," "anywhere," and "else," which contributes to accurate predictions.

where $F$ and $G$ are the (empirical) CDFs of A and B.

Please refer to the paper for further details.

Applying the test on our models with a confidence level (1-$\alpha$) of 0.99 (in other words, significance level ($\alpha$) of 0.01), both tests (1) and (2) resulted in minimum $\epsilon$ values of 0. This indicates that SIR-ABSC and SIR-ABSC (BERT) are stochastically greater than RoBERTa-baseline and BERT-baseline with a confidence level of 0.99, respectively, verifying the superiority of our method.

## F  Additional analysis: does [g] need to be a separate token?

We now compare SIR-ABSC to a variant that does not use a [g] token, but instead uses the target tokens at the end of the input sequence (recall that the input sequence has the form of '[s] sentence [/s] [/s] aspect sequence [/s]'). We call this the SIR-ABSC-[g] variant. As Table 4 shows, the loss in performance is considerable compared to using an independent [g] token as in the original SIR-ABSC. We speculate the drop in performance is due to the original input sentence getting corrupted when we modify the aspect token's attention mask. This indicates the importance of using an additional and independent [g] token for the GNN role as in SIR-ABSC.

## G  SIR-ABSC (BERT) results

We chose RoBERTa as our baseline PLM due to its better overall performance in ABSC tasks (Dai et al., 2021). However, to verify the generality of our method, we also conducted experiments on SIR-ABSC (BERT) which uses BERT instead of RoBERTa. The results are summarized in Table 10. We can see that our method is effective on BERT as well and the significance test results in "stochastic greateness" under significance level of $\alpha = 0.01$, verifying the general effectiveness. The amount of improvement in performance could differ depending on the PLM and we can notice that the amount of enhancement when using RoBERTa (+1.93 on average) is relatively larger than when applied to BERT (+1.60 on average).

| Variable Distance Control (VDC) | Lap14 | | Rest14 | | Twitter | | MAMS | |
|---|---|---|---|---|---|---|---|---|
| | Acc. | F1 | Acc. | F1 | Acc. | F1 | Acc. | F1 |
| *GoBERTA (Position Distance)* | | | | | | | | |
| • *Decreasing-VDC* | 83.4 | 80.4 | 88.5 | 83.2 | 76.5 | 75.3 | 84.7 | 84.3 |
| • VDC = [222211110000] | 83.3 | 80.2 | 88.2 | 82.7 | 76.0 | 74.8 | **84.7** | 84.2 |
| • VDC = [333222111000] | 82.0 | 78.8 | 88.4 | 82.8 | **76.5** | **75.3** | **84.7** | **84.3** |
| • VDC = [444422220000] | **83.4** | **80.4** | 88.4 | 83.1 | 75.6 | 74.4 | **84.7** | 84.2 |
| • VDC = [554433221100] | 83.1 | 79.9 | **88.5** | **83.2** | 75.2 | 73.7 | 84.6 | 84.2 |
| • VDC = [666444222000] | 83.2 | 80.0 | 87.8 | 82.0 | 76.0 | 74.8 | **84.7** | 84.2 |
| • *Constant-VDC* | 83.7 | 80.7 | 88.6 | 83.3 | 76.4 | 75.4 | 84.9 | 84.4 |
| • *Increasing-VDC* | 83.7 | 80.5 | 88.6 | 83.2 | 76.9 | 76.0 | 84.8 | 84.4 |
| • VDC = [000011112222] | 83.5 | 80.3 | 87.8 | 82.1 | **76.9** | **76.0** | 84.7 | 84.3 |
| • VDC = [000111222333] | 83.5 | **80.5** | 88.5 | **83.2** | 75.5 | 74.4 | 84.7 | 84.3 |
| • VDC = [000022224444] | 83.6 | 80.4 | 87.9 | 82.2 | 75.7 | 74.4 | **84.8** | **84.4** |
| • VDC = [001122334455] | 83.3 | 80.3 | **88.6** | 83.1 | 76.1 | 74.9 | **84.8** | **84.4** |
| • VDC = [000222444666] | **83.7** | **80.5** | 88.3 | 82.5 | 76.6 | 75.8 | 84.7 | 84.2 |
| *GoBERTA (Dependency Graph)* | | | | | | | | |
| • *Decreasing-VDC* | 83.7 | 80.6 | **88.2** | 82.8 | 76.5 | 75.4 | 84.8 | 84.3 |
| • VDC = [222211110000] | 83.5 | 80.4 | 88.1 | 82.7 | 75.4 | 74.2 | 84.7 | **84.3** |
| • VDC = [333222111000] | 82.6 | 79.6 | 87.1 | 81.1 | 76.1 | 75.1 | 84.5 | 84.0 |
| • VDC = [444422220000] | 83.4 | 80.5 | 87.9 | 82.2 | **76.5** | **75.4** | **84.8** | **84.3** |
| • VDC = [554433221100] | **83.8** | **80.8** | **88.2** | **82.9** | 75.3 | 74.2 | 84.6 | 84.1 |
| • VDC = [666444222000] | 83.2 | 80.0 | 88.2 | 82.7 | 75.6 | 74.3 | 84.5 | 84.0 |
| • *Constant-VDC* | 83.7 | 80.4 | 88.9 | 83.7 | 76.4 | 75.2 | 85.0 | 84.5 |
| • *Increasing-VDC* | 83.8 | 80.8 | 89.1 | 83.8 | 77.1 | 75.9 | 85.0 | 84.6 |
| • VDC = [000011112222] | **84.1** | **81.1** | 88.3 | 82.8 | **77.2** | **76.3** | 84.6 | 84.2 |
| • VDC = [000111222333] | 83.8 | 80.8 | 89.1 | 83.8 | 76.7 | 75.5 | 84.7 | 84.3 |
| • VDC = [000022224444] | 83.5 | 80.5 | 88.9 | 83.5 | 75.7 | 74.6 | 84.8 | 84.3 |
| • VDC = [001122334455] | 83.2 | 80.2 | **89.3** | **84.1** | 74.7 | 75.9 | **85.0** | **84.6** |
| • VDC = [000222444666] | 82.5 | 79.4 | 88.9 | 83.8 | 76.9 | 75.9 | 84.8 | 84.5 |

Table 8: Detailed experimental results on the effect of VDC. The results show that SIR-ABSC generally shows better performance in the order of decreasing < fixed < increasing VDCs. This result matches our intuition of [g] token imitating GNN as described in Section 3.

| Ablations (Validation) | Lap14 | | Rest14 | | Twitter | | MAMS | |
|---|---|---|---|---|---|---|---|---|
| | Acc. | F1 | Acc. | F1 | Acc. | F1 | Acc. | F1 |
| 1. SIR-ABSC (RoBERTa) | | | | | | | | |
| #1: RoBERTa baseline | 82.1 | 78.9 | 87.6 | 81.7 | 75.6 | 74.5 | 84.4 | 83.9 |
| #2: #1+[g] token+VDC | 84.1 | 81.1 | 89.3 | 84.1 | 77.2 | 76.3 | 85.0 | 84.6 |
| #3 (DAA Ablation #1): #2+DAA (w/o edge label) | 84.2 | 81.3 | 89.3 | 84.3 | 77.2 | 76.0 | 84.8 | 84.3 |
| #4 (DAA Ablation #2): #2+DAA (w/o edge direction) | 84.4 | 81.5 | 89.4 | 84.3 | 77.3 | 76.5 | 84.9 | 84.4 |
| #5 (DAA Ablation #3): #2+DAA (proposed) | 84.6 | 81.8 | 89.4 | 84.4 | 77.2 | 76.3 | 85.2 | 84.7 |
| #6 (SIR-ABSC): #5+Auto-VDC | 85.0 | 82.1 | 89.7 | 84.8 | 77.5 | 76.4 | 85.4 | 84.9 |
| 2. SIR-ABSC (BERT) | | | | | | | | |
| #1: BERT baseline | 79.2 | 75.1 | 85.8 | 79.3 | 76.4 | 74.8 | 84.0 | 82.7 |
| #2: #1+[g] token+VDC | 80.8 | 77.1 | 86.7 | 80.6 | 77.1 | 75.9 | 84.3 | 83.1 |
| #3 (DAA Ablation #1): #2+DAA (w/o edge label) | 81.1 | 77.5 | 86.9 | 80.9 | 77.2 | 75.9 | 84.3 | 83.2 |
| #4 (DAA Ablation #2): #2+DAA (w/o edge direction) | 81.6 | 78.2 | 87.0 | 81.1 | 77.4 | 76.1 | 84.4 | 83.5 |
| #5 (DAA Ablation #3): #2+DAA (proposed) | 81.8 | 78.6 | 87.0 | 81.2 | 77.6 | 76.4 | 84.5 | 83.5 |
| #6 (SIR-ABSC (BERT)): #5+Auto-VDC | 82.2 | 79.1 | 87.3 | 81.6 | 77.7 | 76.6 | 84.7 | 83.6 |

Table 9: A detailed ablation study results.

| SIR-ABSC (BERT) | Lap14 | | Rest14 | | Twitter | | MAMs | |
|---|---|---|---|---|---|---|---|---|
| | Acc. | F1 | Acc. | F1 | Acc. | F1 | Acc. | F1 |
| BERT baseline | 79.2(±0.6) | 75.1(±0.5) | 85.8(±0.7) | 79.3(±0.7) | 76.4(±0.7) | 74.8(±07) | 84.0(±0.3) | 82.7(±0.4) |
| SIR-ABSC (BERT) | 82.2(±0.4) | 79.1(±0.4) | 87.3(±0.7) | 81.6(±0.8) | 77.7(±0.8) | 76.6(±0.7) | 84.6(±0.4) | 83.3(±0.5) |

Table 10: Empirical results on using BERT as the PLM. Note that our technique is effective on BERT as well (significance test resulting in "stochastic greateness") showing comparable results with BERT-based state-of-the-art results. However, the improvement is generally less than when applied to RoBERTa.