# OpenReview forum: "SIR-ABSC: Incorporating Syntax into RoBERTa-based Sentiment Analysis Models with a Special Aggregator Token"
_EMNLP/2023/Conference — EMNLP 2023 Findings_

### Official Review · Reviewer_XaBJ · 2023-08-03

**Soundness:** 2

**Excitement:**

3: Ambivalent: It has merits (e.g., it reports state-of-the-art results, the idea is nice), but there are key weaknesses (e.g., it describes incremental work), and it can significantly benefit from another round of revision. However, I won't object to accepting it if my co-reviewers champion it.

**Justification For Ethical Concerns:**

not involving

**Missing References:**

Inducing target-specific latent structures for aspect sentiment classification.
Discrete opinion tree induction for aspect-based sentiment analysis.
Composition-based Heterogeneous Graph Multi-channel Attention Network for Multi-aspect Multi-sentiment Classification.
and so forth.

**Paper Topic And Main Contributions:**

This work proposes a new way of incorporating dependency information without resorting to GNNs, which is instructive. But the overall presentation is not clear enough, especially Figure 2 cannot show the main idea of ​​the work, which brings troubles to readers. In addition, the proposed DAA module uses the attention mechanism to fuse the labels of the dependency tree, but the subsequent ablation experiments did not sufficiently prove the effectiveness of the method, and no further experiments were conducted to analyze the method. There are also typos, and many illustrations are not clear enough.

**Questions For The Authors:**

1. How effective is DAA?
2. How well does this method work on BERT

**Reasons To Accept:**

It is a new way of incorporating dependency information without resorting to GNNs, which is instructive.

**Reasons To Reject:**

The overall presentation is not clear enough, especially Figure 2 cannot show the main idea of ​​the work, which brings troubles to readers. In addition, the proposed DAA module uses the attention mechanism to fuse the labels of the dependency tree, but the subsequent ablation experiments did not sufficiently prove the effectiveness of the method, and no further experiments were conducted to analyze the method. There are also typos, and many illustrations are not clear enough.

**Reproducibility:**

3: Could reproduce the results with some difficulty. The settings of parameters are underspecified or subjectively determined; the training/evaluation data are not widely available.

**Reviewer Confidence:**

5: Positive that my evaluation is correct. I read the paper very carefully and I am very familiar with related work.

**Typos Grammar Style And Presentation Improvements:**

There are also typos, and many illustrations are not clear enough.  Figure 2 cannot show the main idea of ​​the work, which requires modifying the display.

---

> ### Author Rebuttal · Authors · 2023-08-29
>
> We thank the reviewer XaBJ for the valuable comments and the recognition of our insightful and interesting new idea. We would like to provide more explanations to address your concerns one by one.
>
> [Comment 1] “The overall presentation is not clear enough, especially Figure 2 cannot show the main idea of ​​the work, which brings troubles to readers.”
>
> => We apologize if our paper structure was not concise. In the revised paper, we will update Figure 2 and emphasize the key mechanisms (VDC, DAA, Auto-VDC) for better understanding. In the revised paper, we will also include a bullet list of main contributions at the end of the Introduction section to highlight the differences in our approach more clearly, as below.
>
> “““
>
> Our main contributions are summarized below:
> * We present a novel approach to incorporating syntactic information into PLM with a syntax aggregator token.
> * We highlight two limitations of existing GNN-based approaches (suboptimal interaction and suboptimal aggregation) and provide methods (DAA and Auto-VDC) that effectively alleviate them.
> * To our knowledge, this is the first work to utilize dependency graph information without resorting to GNNs in ABSC.
> * Our model shows state-of-the-art results among RoBERTa-based models on all four widely used ABSC datasets and comparable results on BERT-based models.
>
> “““
>
> [Comment 2] “In addition, the proposed DAA module uses the attention mechanism to fuse the labels of the dependency tree, but the subsequent ablation experiments did not sufficiently prove the effectiveness of the method, and no further experiments were conducted to analyze the method.”
>
> => We have conducted an additional ablation experiment regarding DAA and summarized the results in comment 4.
>
> [Comment 3] There are also typos, and many illustrations are not clear enough.
>
> => Thanks for pointing out typos. We will fix typos in the revised paper. For the illustrations, we’ve included additional figure descriptions for better understanding. Also, we will update Figure 2 in the revised paper.
> The Figure descriptions (captions) will be updated as below:
> * Figure 3: With Auto-VDC, the [g] token automatically learns the ideal VDC configuration. For example, in the lower layers, the [g] token’s attention scores focus on the target aspect ("touch screen functions") and [s] token. In the last layers, the [g] token focuses on the key phrase ``did not enjoy’’ for the final prediction. This shows that Auto-VDC gradually aggregates information from more and more distant nodes in the upper layers, just like heuristic VDCs and GNNs, but in an automatic way, thereby reducing the manual heuristic works.
>
> * Figure 4: A case study on the effect of the Auto-VDC mechanism. The target aspect is “resolution”. Figure (a) shows the tokens corresponding to each VDC value. With Auto-VDC, our model learns to focus more on the key distances (VDCs of 1 and 2) and less on the redundant distance (VDC of 3), as in Figure (b). As a result, the model successfully focuses on the key phrase “has higher resolution” at the last layers instead of concentrating on "fonts are small" which is irrelevant to the target "resolution".
>
>
> [Comment 4] How effective is DAA?
>
> => We used two types of syntactic information: (1) Syntactic distance (i.e., the number of hops from a word to the target aspect based on the dependency graph) and (2) dependency labels (e.g., nsubj). We also tried using POS-tags but observed no significant improvement in performance, thus excluding POS-tags from our final model. Based on your feedback, we conducted an additional analysis on DAA focusing on how the dependency information (edge label and edge direction) affects the model’s performance. We attached the results and summarized our key findings below:
>
> | DAA ablations |      Laptop (Acc/F1-score)      |  Restaurant (Acc/F1-score)|
> |----------|:-------------:|------:|
> | DAA (proposed)|   **84.6/81.8** |  **89.4/84.4** |
> | DAA (w/o edge direction) |    84.4/81.5 |  89.4/84.3 |
> |DAA (w/o edge label) | 84.2/81.3 |   89.3/84.3 |
>
> * [1] DAA (w/o edge direction): we ignore the direction of the edges and assign the embeddings solely based on the dependency label.
> * [2] DAA (w/o edge label): we ignore the edge labels (e.g., nsubj) and assign the embeddings solely based on the edge direction.
> * The results show that the performance decreases for both DAA (w/o edge direction) and DAA (w/o edge label), indicating their importance in DAA performance. Consistent with our intuition, we can observe that the edge label information is relatively more important than edge direction information in DAA.
>
> In addition to the above quantitative analysis, we conducted a qualitative analysis by choosing two examples for case studies. Due to OpenReview not supporting images, we wrote each word and its corresponding attention score next to it in parenthesis. Note that the attention score is the score attended by the [g] token at the PLM's last layer (DAA is only applied to the [g] token).
>
> Case study #1
>
> ---
>
> Text: How pretentious and inappropriate for mj grill to claim that it provides power lunch and **dinners**!
>
> Aspect: dinners
>
> True sentiment: Negative
>
> ```
> dependency tree:
>         lunch--conj-->dinner
>         lunch --cc--> and
>         lunch --compound--> power
>         provides--dobj--> lunch
>         provides--nsubj--> it
>         provides--mark--> that
>         claim--ccomp--> provides
>         claim--aux--> to
>         pretentious--advcl--> claim
>         pretentious--prep--> for
>         for --pobj-->grill
>         grill--compound-->mj
>         pretentious--conj--> inappropriate
>         pretentious--cc--> and
>         pretentious--advmod--> how
> ```
>
> ---
>
> Without DAA:
> * [g] token's attention scores:
>     * [s] (0.0001) [g] (0.0184)  how (0.0039) pretentious (0.0008) and (0.0005) inappropriate (0.0004) for (0.0040) mj (X) grill (X) to (0.002) claim (0.0158) that (0.0222) it (0.0521) provides (0.1365) power (0.1476) lunch (0.1782) and (0.3286) dinners (0.0371) ! (0.0518)
> * Prediction:
>     * Neutral ([Negative: 0.0101, Neutral: 0.7712, Positive: 0.2188])
>
> With DAA:
> * [g] token's attention scores (attentions scores with more than x10 increase are shown in bold case):
>     * **[s] (0.0022)** [g] (0.0614)  how (0.00391) **pretentious (0.0556)** **and (0.0150)** **inappropriate (0.0507)** for (0.0244) mj (X) grill (X) **to (0.0620)** claim (0.0590) that (0.0773) it (0.1067) provides (0.2374) power (0.0858) lunch (0.0165) and (0.0928) dinners (0.0008) ! (0.0132)
> * Prediction:
>     * Negative ([Negative: 0.7758, Neutral: 0.1160, Positive: 0.1082])
>
> Case study #2
>
> ---
>
> Text: Anywhere else, the **prices** would be 3x high
>
> Aspect: prices
>
> True sentiment: Positive
>
> ```
> dependency tree:
>         be--aux-->would
>         be --attr--> 3x
>         3x --advmod--> high
>         high--advmod--> as
>         be --nsubj--> prices
>         be --advmod--> else
>         prices--det--> the
>         else --advmod--> anywhere
>
> ```
>
> ---
>
> Without DAA:
> * [g] token's attention scores:
>     * [s] (0.0023) [g] (0.045) anywhere (0.0013) else (0.0032) , (0.0116) the (0.1280) prices (0.0320) would (0.1441) be (0.0887) 3 (0.0552) x (0.123) high (0.2666) ! (0.099)
> * Prediction:
>     * Negative ([Negative: 0.9970,  Neutral: 0.0021,  Positivie: 0.0009])
>
> With DAA:
> * [g] token's attention scores (attentions scores with more than x10 increase are shown in bold case):
>     * [s] (0.0049) [g] (0.094) **anywhere (0.0401) else (0.0678)**, (0.0316) the (0.0908) prices (0.0444) would (0.1338) be (0.1107) 3 (0.0604) x (0.094) high (0.1059) ! (0.1442)
> * Prediction:
>     * Positive ([Negative: 0.3558,  Neutral: 0.0021,  Positive: 0.6421])
>
> From the above examples, we can observe that the attention scores of words (pretentious, and, inappropriate) and (anywhere, else) get much higher when DAA is applied. We speculate that this is because the DAA learns the importance of "advcl" and "advmod" labels and therefore assign high attention scores on (pretentious--advcl--> claim, be --advmod--> else,  and else --advmod--> anywhere). This is also consistent with our intuition since adverbial modifiers are related to adjectives which is highly likely to be important for deciding the sentiment.
>
> [Comment 5] How well does this method work on BERT?
>
> ⇒ We conducted experiments using BERT (SIR-ABSC (BERT)) instead of RoBERTa (We already included the experiment in Lines 417 - 421). Our empirical results on using BERT as the PLM (Appendix F) show our method is effective on BERT as well (significant test resulting in ``stochastic greatness”).
>
> To address your comment, we’ve conducted additional ablation experiments on the effectiveness of DAA on SIR-ABSC (BERT) (Will be added in Appendix F). As shown in the below table, we can observe that each method (DAA and Auto-VDC) improves the performance over the BERT baseline. These results indicate that leveraging dependency labels (DAA) and a mechanism to learn to focus on specific distances based on the input (Auto-VDC) effectively work to enhance model performance.
>
> | SIR-ABSC (BERT) ablations |      Laptop (Acc/F1-score)      |  Restaurant (Acc/F1-score)|
> |----------|:-------------:|------:|
> | BERT baseline |   79.2/75.1 | 85.8/79.3 |
> |   $\hspace{5pt}$  + DAA |    81.8/78.6 | 87.0/81.2 |
> |  $\hspace{5pt}$ + Auto-VDC (=SIR-ABSC (BERT))| **82.2/79.1** |   **87.3/81.6** |
>
> Due to the time limit for rebuttal, we were only able to conduct the ablation experiments for SIR-ABSC (BERT) on the Laptop and Restaurant14 dataset. Experiments on the remaining datasets (MAMS and Twitter) will be included in the revised paper.
>
> [Comment 6] Missing References: Inducing target-specific latent structures for aspect sentiment classification. Discrete opinion tree induction for aspect-based sentiment analysis. Composition-based Heterogeneous Graph Multi-channel Attention Network for Multi-aspect Multi-sentiment Classification. and so forth.
>
> ⇒ Thanks for your suggestions on references. We have summarized the main contributions of paper [1], [2], [3] below:
> * Paper [1] is to amalgamate a dependency tree and a target-specific latent graph via a GCN gating mechanism. The primary focus lies in inducing latent structures for aspect-level sentiment classification. They establish individual latent graphs for each aspect by leveraging the Kumaraswamy distribution alongside a multi-head attention approach. These graphs are then merged using the same graph convolutional matrix, thereby avoiding the introduction of extra parameters within the gating mechanism.
> * Similarly, Paper [2] induces a discrete opinion tree structure for each aspect term. They train the attention-based tree inductor through a policy network and regularize attention weights using KL divergence. This regularization prompts the aspect term to attend contexts with shorter distances.
> * Paper [3] proposes a multi-channel aggregating mechanism (similar to multi-view learning) to utilize the syntactic dependency graphs. The approach involves a three-view aggregation strategy: (1) original view: original node embeddings initialized with the output of BERT, (2) node type view: distinguishes between aspect nodes and the rest (referred to as context nodes), and (3) edge type view: uses a composed dependency tag for aspect-context nodes and a sum of intermediate tokens along the dependency graph for aspect-aspect nodes.
>
> The main difference between our work with these references is synchronicity. Existing GNN-based models in ABSC (including the above references) operate asynchronously, where GNN operates after PLM finishes operating. The core idea of our work is to remove the GNN module, which inevitably causes this asynchronous behavior. Instead, we inserted a novel aggregator token, the [g] token, inside the PLM. By manipulating the attention mask of the [g] token, we successfully replaced the work of GNN and presented a fully integrated and synchronous model for ABSC. To our knowledge, this is the first work to utilize the dependency graph information without any GNN modules in ABSC.
>
> From the syntax information perspective, papers [1] and [2] utilize undirected graphs. While paper [3] and our work both incorporate trainable embeddings for edge information, the consideration of edge direction information is absent in paper [3]. To effectively leverage both edge label and direction information, we introduce the DAA mechanism. In the context of DAA's ablation studies (comments 4 and 5), the utilization of edge label and direction information demonstrably enhances our model's performance.
>
> We will include the above papers and discussions in Section 2.2 of our revised paper.
>
> We hope these revisions will make our paper more robust. Thank you again for your thoughtful feedback.
>
> ---
>
> Reference:
>
> * [1] Chenhua Chen, Zhiyang Teng, and Yue Zhang. 2020. Inducing Target-Specific Latent Structures for Aspect Sentiment Classification. In Proceedings of the 2020 Conference on Empirical Methods in Natural Language Processing (EMNLP), pages 5596–5607, Online. Association for Computational Linguistics.
> * [2] Chenhua Chen, Zhiyang Teng, Zhongqing Wang, and Yue Zhang. 2022. Discrete Opinion Tree Induction for Aspect-based Sentiment Analysis. In Proceedings of the 60th Annual Meeting of the Association for Computational Linguistics (Volume 1: Long Papers), pages 2051–2064, Dublin, Ireland. Association for Computational Linguistics.
> * [3] Hao Niu, Yun Xiong, Jian Gao, Zhongchen Miao, Xiaosu Wang, Hongrun Ren, Yao Zhang, and Yangyong Zhu. 2022. Composition-based Heterogeneous Graph Multi-channel Attention Network for Multi-aspect Multi-sentiment Classification. In Proceedings of the 29th International Conference on Computational Linguistics, pages 6827–6836, Gyeongju, Republic of Korea. International Committee on Computational Linguistics.

---

### Official Review · Reviewer_hL41 · 2023-08-05

**Soundness:** 4

**Excitement:**

4: Strong: This paper deepens the understanding of some phenomenon or lowers the barriers to an existing research direction.

**Missing References:**

The proposed idea is similar to the following paper. The authors should discuss the difference.
https://ieeexplore.ieee.org/document/9976197

**Paper Topic And Main Contributions:**

This paper presents Syntax-Integrated RoBERTa for ABSC (SIR-ABSC) that incorporates syntax directly into the language model by using a novel aggregator token.

**Questions For The Authors:**

1. What kinds of syntatic information are used? How do they affect the performance? It is encouraged to provide a detailed analysis in the experiments.
2. The contribution should be highlighted at the end of Introduction.

**Reasons To Accept:**

This paper presents a new method to address previous two limitations: suboptimal interaction and suboptimal aggregation for the ABSC task.

**Reasons To Reject:**

None

**Reproducibility:**

4: Could mostly reproduce the results, but there may be some variation because of sample variance or minor variations in their interpretation of the protocol or method.

**Reviewer Confidence:**

4: Quite sure. I tried to check the important points carefully. It's unlikely, though conceivable, that I missed something that should affect my ratings.

---

> ### Author Rebuttal · Authors · 2023-08-29
>
> We would like to thank reviewer hL41 for the detailed and thoughtful review. We are delighted to hear that you found our work interesting. Each comment and how we have addressed them in our revised paper is summarized below.
>
> [Comment 1] What kinds of syntactic information are used? How do they affect the performance? It is encouraged to provide a detailed analysis in the experiments.
>
> => We used two types of syntactic information: (1) Syntactic distance (i.e., the number of hops from a word to the target aspect based on the dependency graph) and (2) dependency labels (e.g., nsubj). We also tried using POS-tags but observed no significant improvement in performance, thus excluding POS-tags from our final model. Based on your feedback, we conducted an additional analysis on DAA focusing on how the dependency information (edge label and edge direction) affects the model’s performance. We attached the results and summarized our key findings below:
>
> | DAA ablations |      Laptop (Acc/F1-score)      |  Restaurant (Acc/F1-score)|
> |----------|:-------------:|------:|
> | DAA (proposed)|   **84.6/81.8** |  **89.4/84.4** |
> | DAA (w/o edge direction) |    84.4/81.5 |  89.4/84.3 |
> |DAA (w/o edge label) | 84.2/81.3 |   89.3/84.3 |
>
> * [1] DAA (w/o edge direction): we ignore the direction of the edges and assign the embeddings solely based on the dependency label.
> * [2] DAA (w/o edge label): we ignore the edge labels (e.g., nsubj) and assign the embeddings solely based on the edge direction.
> * The results show that the performance decreases for both DAA (w/o edge direction) and DAA (w/o edge label), indicating their importance in DAA performance. Consistent with our intuition, we can observe that the edge label information is relatively more important than edge direction information in DAA.
>
> In addition to the above quantitative analysis, we conducted a qualitative analysis by choosing two examples for case studies. Due to OpenReview not supporting images, we wrote each word and its corresponding attention score next to it in parenthesis. Note that the attention score is the score attended by the [g] token at the PLM's last layer (DAA is only applied to the [g] token).
>
> Case study #1
>
> ---
>
> Text: How pretentious and inappropriate for mj grill to claim that it provides power lunch and **dinners**!
>
> Aspect: dinners
>
> True sentiment: Negative
>
> ```
> dependency tree:
>         lunch--conj-->dinner
>         lunch --cc--> and
>         lunch --compound--> power
>         provides--dobj--> lunch
>         provides--nsubj--> it
>         provides--mark--> that
>         claim--ccomp--> provides
>         claim--aux--> to
>         pretentious--advcl--> claim
>         pretentious--prep--> for
>         for --pobj-->grill
>         grill--compound-->mj
>         pretentious--conj--> inappropriate
>         pretentious--cc--> and
>         pretentious--advmod--> how
> ```
>
> ---
>
> Without DAA:
> * [g] token's attention scores:
>     * [s] (0.0001) [g] (0.0184)  how (0.0039) pretentious (0.0008) and (0.0005) inappropriate (0.0004) for (0.0040) mj (X) grill (X) to (0.002) claim (0.0158) that (0.0222) it (0.0521) provides (0.1365) power (0.1476) lunch (0.1782) and (0.3286) dinners (0.0371) ! (0.0518)
> * Prediction:
>     * Neutral ([Negative: 0.0101, Neutral: 0.7712, Positive: 0.2188])
>
> With DAA:
> * [g] token's attention scores (attentions scores with more than x10 increase are shown in bold case):
>     * **[s] (0.0022)** [g] (0.0614)  how (0.00391) **pretentious (0.0556)** **and (0.0150)** **inappropriate (0.0507)** for (0.0244) mj (X) grill (X) **to (0.0620)** claim (0.0590) that (0.0773) it (0.1067) provides (0.2374) power (0.0858) lunch (0.0165) and (0.0928) dinners (0.0008) ! (0.0132)
> * Prediction:
>     * Negative ([Negative: 0.7758, Neutral: 0.1160, Positive: 0.1082])
>
> Case study #2
>
> ---
>
> Text: Anywhere else, the **prices** would be 3x high
>
> Aspect: prices
>
> True sentiment: Positive
>
> ```
> dependency tree:
>         be--aux-->would
>         be --attr--> 3x
>         3x --advmod--> high
>         high--advmod--> as
>         be --nsubj--> prices
>         be --advmod--> else
>         prices--det--> the
>         else --advmod--> anywhere
>
> ```
>
> ---
>
> Without DAA:
> * [g] token's attention scores:
>     * [s] (0.0023) [g] (0.045) anywhere (0.0013) else (0.0032) , (0.0116) the (0.1280) prices (0.0320) would (0.1441) be (0.0887) 3 (0.0552) x (0.123) high (0.2666) ! (0.099)
> * Prediction:
>     * Negative ([Negative: 0.9970,  Neutral: 0.0021,  Positivie: 0.0009])
>
> With DAA:
> * [g] token's attention scores (attentions scores with more than x10 increase are shown in bold case):
>     * [s] (0.0049) [g] (0.094) **anywhere (0.0401) else (0.0678)**, (0.0316) the (0.0908) prices (0.0444) would (0.1338) be (0.1107) 3 (0.0604) x (0.094) high (0.1059) ! (0.1442)
> * Prediction:
>     * Positive ([Negative: 0.3558,  Neutral: 0.0021,  Positive: 0.6421])
>
> From the above examples, we can observe that the attention scores of words (pretentious, and, inappropriate) and (anywhere, else) get much higher when DAA is applied. We speculate that this is because the DAA learns the importance of "advcl" and "advmod" labels and therefore assign high attention scores on (pretentious--advcl--> claim, be --advmod--> else,  and else --advmod--> anywhere). This is also consistent with our intuition since adverbial modifiers are related to adjectives which is highly likely to be important for deciding the sentiment.
>
> [Comment 2] The contribution should be highlighted at the end of Introduction.
>
> => Following your advice, we will add a bullet list of main contributions at the end of the introduction:
>
> “““
>
> Our main contributions are summarized below:
> * We present a novel approach to incorporating syntactic information into PLM with a syntax aggregator token.
> * We highlight two limitations of existing GNN-based approaches (suboptimal interaction and suboptimal aggregation) and provide methods (DAA and Auto-VDC) that effectively alleviate them.
> * To our knowledge, this is the first work to utilize dependency graph information without resorting to GNNs in ABSC.
> * Our model shows state-of-the-art results among RoBERTa-based models on all four widely used ABSC datasets and comparable results on BERT-based models.
>
> “““
>
> [Comment 3] Missing References: The proposed idea is similar to the following paper. The authors should discuss the difference. https://ieeexplore.ieee.org/document/9976197
>
> => We will add the corresponding paper (SGAN) in Section 2.2. The main difference between our work and SGAN is the synchronicity. SGAN, similar to other general GNN-based methods, operates asynchronously (i.e., BiLSTM operates first and then the SGAN module), limiting their interaction, which is the central limitation we wanted to tackle in this paper. SIR-ABSC, on the other hand, is a synchronous model where the semantic and syntactic aggregator modules get updated simultaneously via the newly introduced [g] token inside the PLM. The DAA mechanism is also slightly different from SGAN. SGAN proposes a unique way that first expands dependency labels using the POS-tag of the source node and then uses a target embedding-oriented weighted sum for aggregating multi-head vectors. SIR-ABSC, on the other hand, does not use the POS-tag information and it reuses the multi-head self-attention mechanism in the PLM for utilizing the dependency labels.
>
> We hope these revisions will make our paper more robust. Thank you again for your thoughtful feedback.

---

### Official Review · Reviewer_HrVu · 2023-08-08

**Soundness:** 2

**Excitement:**

2: Mediocre: This paper makes marginal contributions (vs non-contemporaneous work), so I would rather not see it in the conference.

**Paper Topic And Main Contributions:**

This paper present a simple, but effective method to incorporate syntactic dependency information directly into transformer-based language models for ABSC. Experiment results show effective of the proposed model.

**Reasons To Accept:**

The proposed model is novel, interest, and simple. Experiment results show effective of the proposed model.

**Reasons To Reject:**

There are already many researches focus on employing graph-based models for ABSC, what is the difference between the proposed model and previous studies.

The authors should compare the proposed model with previous studies.

**Reproducibility:**

3: Could reproduce the results with some difficulty. The settings of parameters are underspecified or subjectively determined; the training/evaluation data are not widely available.

**Reviewer Confidence:**

4: Quite sure. I tried to check the important points carefully. It's unlikely, though conceivable, that I missed something that should affect my ratings.

---

> ### Author Rebuttal · Authors · 2023-08-29
>
> We would like to thank Reviewer HrVu for the insightful comments. We are glad to hear that you found our work novel and interesting. We summarized how we have addressed your comments in our revised paper below. We hope these revisions make our paper stronger.
>
> ---
>
> [Comment 1] “There are already many researches focus on employing graph-based models for ABSC, what is the difference between the proposed model and previous studies.”
>
> =>  This paper presents a new approach to incorporating syntactic information into PLMs using a new syntax-aggregator token ([g] token) instead of resorting to GNNs. To our knowledge, this is the first work to utilize dependency graph information without using a GNN in ABSC. We summarized the main advantages of SIR-ABSC over standard GNN-based methods below.
>
>
> 1. Since the aggregator token is part of the input sequence, it can naturally make use of the existing modules in the PLM. Specifically,
>     * **Reusing existing parameters:** The [g] token can take advantage of the existing and pre-trained self-attention parameters in the PLM. Therefore, it can effectively aggregate syntactic information with only a minor additional work of manipulating [g] token’s attention mask (explained in Lines 271-274).
>     * **Fully-integratedness:** [s] (=semantic aggregator) and [g] (=syntactic aggregator) tokens interact with each other throughout the entire 12 layers of PLM, making them synchronously updated compared to the asynchronous behavior of standard GNN-based approaches (i.e., PLM operates first and then the GNN operates). Through ablation studies, we show that this synchronicity (or fully-integratedness) is crucial for SIR-ABSC’s performance, which is absent in standard GNN-based methods (explained in the section “How does SIR-ABSC alleviate the suboptimal interaction issue?” (Lines 474-494) and Table 4 variant 1).
>     * **Utilizing [s] token’s inherent strength:** We hypothesize that the [s] token implicitly learns a meaningful ability to summarize information from the input sentence as an aggregator during the heavy pre-training process. The [g] token can easily benefit from this inherent strength of the [s] token by initializing [g]’s embedding with [s]’s embedding, leading to a better performance (explained in the section “Inherent strength of the pre-trained [s] token as an aggregator” (Lines 520-535) and Table 4 variant 2).
>
>
> 2. The DAA mechanism allows SIR-ABSC to utilize the dependency label information, which contains useful information for the ABSC task (explained in Lines 318-338, 495-506).
> 3. The Auto-VDC mechanism allows SIR-ABSC to adjust and focus more on the key syntactic distance of the input data, a substantial advantage over previous GNN-based models (explained in section “How does SIR-ABSC alleviate the suboptimal aggregation issue?” (Lines 495-519) and Figure 4).
>
> In the revised paper, we will include a bullet list of main contributions at the end of the Introduction section to highlight the differences in our approach more clearly, as below.
>
> """
>
> Our main contributions are summarized below:
> * We present a novel approach to incorporating syntactic information into PLM with a syntax aggregator token.
> * We highlight two limitations of existing GNN-based approaches (suboptimal interaction and suboptimal aggregation) and provide methods (DAA and Auto-VDC) that effectively alleviate them.
> * To our knowledge, this is the first work to utilize dependency graph information without resorting to GNNs in ABSC.
> * Our model shows state-of-the-art results among RoBERTa-based models on all four widely used ABSC datasets and comparable results on BERT-based models.
>
> """
>
> [Comment 2] The authors should compare the proposed model with previous studies.
>
> => We compared with previous studies in Section 2.2 (Combining PLMs with syntax), but we apologize if the comparison was not exhaustive enough. Following your advice, we will include more details in Section 2.2 by including more references ([1], [2], [3], [4]).
>
> We truly appreciate your valuable reviews and feedback.
>
> ---
>
> Reference:
> * [1] Li Yuan, Jin Wang, Liang-Chih Yu, Xuejie Zhang. 2020. Syntactic Graph Attention Network for Aspect-Level Sentiment Analysis, In IEEE Transactions on Artificial Intelligence, pages 1-15,
> * [2] Chenhua Chen, Zhiyang Teng, and Yue Zhang. 2020. Inducing Target-Specific Latent Structures for Aspect Sentiment Classification. In Proceedings of the 2020 Conference on Empirical Methods in Natural Language Processing (EMNLP), pages 5596–5607, Online. Association for Computational Linguistics.
> * [3] Chenhua Chen, Zhiyang Teng, Zhongqing Wang, and Yue Zhang. 2022. Discrete Opinion Tree Induction for Aspect-based Sentiment Analysis. In Proceedings of the 60th Annual Meeting of the Association for Computational Linguistics (Volume 1: Long Papers), pages 2051–2064, Dublin, Ireland. Association for Computational Linguistics.
> * [4] Hao Niu, Yun Xiong, Jian Gao, Zhongchen Miao, Xiaosu Wang, Hongrun Ren, Yao Zhang, and Yangyong Zhu. 2022. Composition-based Heterogeneous Graph Multi-channel Attention Network for Multi-aspect Multi-sentiment Classification. In Proceedings of the 29th International Conference on Computational Linguistics, pages 6827–6836, Gyeongju, Republic of Korea. International Committee on Computational Linguistics.

---

### Meta-Review · Area_Chair_ep5F · 2023-09-20

**Recommendation:** 3

**Metareview:**

This paper proposes a new method that incorporates syntactic dependency information into a Transformer model without using GNNs for Aspect-Based Sentiment Classification (ABSC) task. Concretely, the proposed method inserts a new token [g] to represent syntactic information. Moreover, the authors introduce VDC (Variable distance control) that specifies the attention mask used by [g], and further extend it from heuristic to learnable. They also introduce DAA (dependency-aware aggregation) to represent dependency labels.

As reviewers mentioned, the authors present a new and interesting method for ABSC. Unlike the baselines, this method does not use GNNs therefore mitigates suboptimal interaction and aggregation for the task.

I appreciated that the authors were highly engaged in the discussion period and provided a large amount of additional results and clarification. Addressing reviewers' feedback, the authors conducted ablation experiments to show the impact of DAA. Although DAA improves the results only by a small margin, when combined with Auto-VDC the model results in the best performance. Additionally, the authors provided another set of comparisons with the BERT-based baselines. Reflecting on the authors' comment, their method is only competitive in this setting, however, methodological novelty and achieving high results with Roberta model is an important contribution. Finally, I believe incorporating additional content provided during the discussion period, will improve the paper.

---

### Decision · Program_Chairs · 2023-10-07

**Decision:**

Accept-Findings

**Comment:**

This paper proposes a new method that incorporates syntactic dependency information into a Transformer model without using GNNs for Aspect-Based Sentiment Classification (ABSC) task. Concretely, the proposed method inserts a new token [g] to represent syntactic information. Moreover, the authors introduce VDC (Variable distance control) that specifies the attention mask used by [g], and further extend it from heuristic to learnable. They also introduce DAA (dependency-aware aggregation) to represent dependency labels.

As reviewers mentioned, the authors present a new and interesting method for ABSC. Unlike the baselines, this method does not use GNNs therefore mitigates suboptimal interaction and aggregation for the task.

I appreciated that the authors were highly engaged in the discussion period and provided a large amount of additional results and clarification. Addressing reviewers' feedback, the authors conducted ablation experiments to show the impact of DAA. Although DAA improves the results only by a small margin, when combined with Auto-VDC the model results in the best performance. Additionally, the authors provided another set of comparisons with the BERT-based baselines. Reflecting on the authors' comment, their method is only competitive in this setting, however, methodological novelty and achieving high results with Roberta model is an important contribution. Finally, I believe incorporating additional content provided during the discussion period, will improve the paper.